# In Vitro Evaluation of the Interaction of Seven Biologically Active Components in *Anemarrhenae rhizoma* with P-gp

**DOI:** 10.3390/molecules27238556

**Published:** 2022-12-05

**Authors:** Jianying Dai, Yuzhen He, Jiahao Fang, Hui Wang, Liang Chao, Liang Zhao, Zhanying Hong, Yifeng Chai

**Affiliations:** 1School of Pharmacy, Naval Medical University, Shanghai 200433, China; 2Shanghai Key Laboratory for Pharmaceutical Metabolite Research, Shanghai 200433, China; 3Department of Pharmacy, Shanghai Baoshan Luodian Hospital, Shanghai 201908, China

**Keywords:** *Anemarrhenae rhizoma*, P-gp, transmembrane transport, MDCK-MDR1 cells, interaction

## Abstract

The efficacy and pharmacokinetics of the biologically active components in *Anemarrhenae rhizoma* (AR) would be affected by the interaction of P-glycoprotein(P-gp) and effective components in AR. However, little is known about the interaction between them. The goal of this research was to examine the transmembrane absorption of timosaponin AIII(TAIII), timosaponin BII(TBII), sarsasapogenin (SSG), mangiferin(MGF), neomangiferin(NMGF), isomangiferin(IMGF), and baohuosideI(BHI) in AR and their interaction with P-gp. Seven effective components in AR(TAIII, TBII, SSG, MGF, NMGF, IMGF, and BHI) were investigated, and MDCK-MDR1 cells were used as the transport cell model. CCK-8 assays, bidirectional transport assays, and Rhodamine-123 (Rh-123) transport assays were determined in the MDCK-MDR1 cells. LC/MS was applied to the quantitative analysis of TAIII, TBII, MGF, NMGF, IMGF, SSG, and BHI in transport samples. The efflux ratio of MGF, TAIII, TBII, and BHI was greater than 2 and significantly descended with the co-administration of Verapamil, indicating MGF, TAIII, TBII, and BHI as the substrates of P-gp. The efflux ratio of the seven effective components in the extracts (10 mg/mL) of AR decreased from 3.00~1.08 to 1.92~0.48. Compared to the efflux ratio of Rh-123 in the control group (2.46), the efflux ratios of Rh-123 were 1.22, 1.27, 1.25, 1.09, 1.31, and 1.47 by the addition of TAIII, TBII, MGF, IMGF, NMGF, and BHI, respectively, while the efflux ratio of Rh-123 with the co-administration of SSG had no statistical difference compared to the control group. These results indicated that MGF, TAIII, TBII, and BHI could be the substrates of P-gp. TAIII, TBII, MGF, IMGF, NMGF, and BHI show the effect of inhibiting P-gp function, respectively. These findings provide important basic pharmacological data to assist the therapeutic development of AR constituents and extracts.

## 1. Introduction

*Anemarrhenae rhizoma* (AR) is the dried rhizome of *Anemarrhena asphodeloides Bunge*. AR has traditionally been used to treat fever, cough, and diabetes. Currently, pharmacological research on AR is developing. AR can be used to treat a variety of diseases, including Alzheimer’s disease, Parkinson’s disease, and schizophrenia [1]. Steroid saponins, flavonoids, lignin, and polysaccharides are the main components of AR, with steroidal saponins and flavonoids being the most pharmacologically effective ones. [2]. The main monomer saponins, aglycon and flavonoid in AR are timosaponin A I~A IV (TAI~IV), BI~BVI (TBI~VI), sarsasapogenin (SSG), markogenin, smilagenin, mangiferin (MGF), neomangiferin (NMGF), isomangiferin (IMGF), baohuoside I (BHI), neogitogenin, officinalisinin and so on [3]. 

Numerous pharmacological research on the aforementioned components of AR have been conducted recently. SSG can promote apoptosis in gastric cancer cells BGC-823 and HepG2 [4,5]. TAIII can induce the apoptosis of breast cancer cells, HeLa cells, colon cancer cells, melanoma cells, brain glioma cells, and pancreatic cancer cells [6,7,8]. TBII and TAIII possess memory-improving and ameliorate learning deficits in mice [9,10]. MGF can effectively stimulate the choline receptor and ameliorate long-term memory problems in Alzheimer’s model mice [11]. NMGF has the ability to ameliorate murine calvarial inflammatory osteolysis and inflammation of benign prostatic hyperplasia [12,13]. IMGF has the therapeutic potential to combat viral infections [14,15]. BHI has several functions, including anti-osteoporosis, cognitive dysfunction improvement, cerebral ischemia-reperfusion injury protection, neuroprotection, and other functions [16]. Nonetheless, low bioavailability is a common issue with the effective components in AR as other botanicals. Another hidden danger is the possibility of drug-drug interactions in herbal therapy for effective ingredients in AR. In addition to that, to our knowledge, few previous studies have reported the absorption of steroidal saponins and flavonoids from AR.

P-glycoprotein (P-gp), a 170k Da cell membrane protein encoded by the MDR-1 gene in humans [17], is considered the most contributing efflux transporter among the ATP binding cassette(ABC) transporter family. P-gp is the main efflux pump with a broad substrate spectrum and is expressed in cells of various biological barriers, e.g., the placental, blood-testis, blood–brain, and blood–ocular barriers, which impact the efficacy of many chemotherapeutic and neurocentral drugs. Many biologically active components in AR have been known for their high affinity and specificity with P-gp and have the potential to be P-gp inhibitors or modulators. MGF’s low bioavailability is a major impediment to its further development, and P-gp inhibition slightly enhanced MGF exposure [18]. Pharmacokinetic experiments and rat intestinal perfusion model in situ found that P-gp could affect TAIII absorption, and this effect can be eliminated by the combination of P-gp inhibitors [19]. Due to the low toxicity profile and high specificity of natural products, research on finding P-gp inhibitors is becoming a point of interest for modern researchers [20]. Many natural products have been known to inhibit P-gp, such as *Euphorbia intisy* essential oil [21], caffeic acid [22], and Taxifolin [23]. Many of the biological components in AR also have the potential to become P-gp inhibitors. However, the interaction between P-gp and the effective components of AR remains unclear. This is also an urgent problem to be solved in the research and development process of AR. 

The Madin-Darby canine kidney (MDCK) cell line is a kind of tightly junction cell line developed from dog renal epithelial cells with low levels of transporter expression and low metabolic activity. In terms of genetics, lipid, and protein content, MDCK cells are the most suitable epithelial cell line. Pastan et al. [24] transfected MDCK cells with the human MDR-1 gene to establish a P-gp over-expressed cell line called the MDCK-MDR1 cell. The short culture cycle, homogeneity across generations, and strong expression of human P-gp are all benefits of MDCK-MDR1 cells [25]. In general, research in MDCK-MDR1 cells involves a bidirectional investigation of the transport of drugs. Results obtained for apical–basolateral (A to B) and basolateral–apical (B to A) directions may provide an idea regarding probable substrates for P-gp through analysis of the ratio B-A/A-B [26]. This method is widely used to evaluate the transport characteristics of the effective components of AR in this study. 

The goal of this research was to examine the transmembrane absorption of TAIII, TBII, SSG, MGF, NMGF, IMGF, and BHI in AR and their interaction with P-gp. The chemical structures of these compounds are shown in Figure 1. We also extracted AR powder with water and ethanol, then examined the interaction with the P-gp of each compound in the AR extract. Bidirectional transport assays and R-123 transport assays were used to investigate the interaction between seven effective components in AR and P-gp in MDCK-MDR1 cells, and a quantitative analysis method for these components was established.

## 2. Results

### 2.1. Identification of Seven Components in AR as P-gp Substrates

The highest concentration under IC_50_ of MDCK-MDR1 cells were 2 μM for TAIII, 1.6 μM for TBII, 4.7 μM for MGF, 4.7 μM for IMGF, 3.4 μM for NMGF, 1.2 μM for SSG, and 3.9 μM for BHI by the CCK-8 assay, respectively (n = 3). The study indicated that the cytotoxicity of the seven components at the above concentration was minimal, and it was suitable for subsequent transporter experiments.

The P_app_ and ER were calculated from the concentrations of the effective constituents of AR in the samples at different time points (30, 60, 90, and 120 min) in the bidirectional transport experiments and compared to respective permeability in the cells co-administrated with VER. The results are shown in Table 1. The efflux ratio of TAIII, TBII, MGF, and BHI were 2.26, 2.32, 2.23, and 3.00, respectively, which were greater than the generally accepted thresholds of 2 for substrates of P-gp. Moreover, the transport of TAIII, TBII, MGF, and BHI were significantly inhibited by Verapamil, a P-gp inhibitor. The ER of TAIII, TBII, MGF, and BHI decreased from 2.26, 2.32, 2.23, and 3.00 to 0.88, 0.65, 0.53, and 1.08, respectively, as shown in Table 1 and Figure 2. The results indicated that all four compounds were potential substrates of P-gp. 

### 2.2. Transport Characteristics of Seven Components in AR Extract

To investigate the highest concentration of AR extract under IC_50_ of MDCK-MDR1 cell, a CCK-8 assay was performed. The highest concentration under IC_50_ of MDCK-MDR1 cells was 0.8 μg/mL for AR extract (n = 3). Follow-up bidirectional transport experiments were conducted using the above concentration.

The P_app_ and ER of seven compounds were calculated by the results of AR extract bidirectional transport experiments and summarized in Table 2. Figure 3 presents the comparison of the efflux ratio of seven components in the different administrations of AR extracts and monomers. As shown in Table 3, the P_app (AP-BL)_ and P_app (BL-AP)_ of MGF, IMGF, and NMGF decreased in the bidirectional transport assay of the AR extract, and the ER of MGF, NMGF, and IMGF decreased from 2.23, 1.13, and 1.08 to 0.48, 0.67 and 0.91, respectively. The P_app (AP-BL)_ and P_app (BL-AP)_ of TAIII increased, and the ER of TAIII decreased from 2.26 to 0.63, on the contrary. The P_app (AP-BL)_ of TBII and SGG increased while the P_app (BL-AP)_ decreased, and the ER of TBII and SSG decreased from 2.32 and 1.51 to 0.83 to 0.87. The P_app (AP-BL)_ and P_app (BL-AP)_ of BHI increased I, and the ER of BHI decreased from 2.26 to 0.63 as well. The transport characteristics of seven effective constituents in the AR extract differed from those in monomers, suggesting that these seven components are either P-gp substrates or P-gp inhibitors that affect P-gp function.

### 2.3. Identification of AR Components as Potential P-gp Inhibitors

According to the results depicted in Table 3, six of the seven components of AR, i.e., TAIII, TBII, MGF, IMGF, NMGF, and BHI, tested induced a marked decrease on the efflux ratio of Rh123 in MDCK-MDR1 cells. The results of the above six compounds showed statistically significant differences between treated cells and untreated control cells (* *p* < 0.05). Specifically, IMGF at 4.7 μM (ER = 1.09) produced decreases in the efflux of Rh123 similar to that exhibited by valspodar at 0.625μM (positive control as P-gp inhibitor). Compared with the control group (ER = 2.46), TAIII, TBII, MGF, IMGF, NMGF, and BHI decreased the ER of Rh-123 to 1.22, 1.27, 1.25, 1.31, and 1.47, respectively. 

### 2.4. Molecular Docking of TAIII, TBII, MGF, and BHI to P-gp

Based on our above findings, TAIII, TBII, MGF, and BHI were both the substrates and inhibitors of P-gp. In silico molecular dockings were thus performed to investigate the binding of TAIII, TBII, MGF, and BHI with P-gp. The binding energy of TAIII, TBII, MGF, BHI, elacridar, and verapamil with transmembrane domain and nucleotide-binding domain of P-gp predicted on the basis of the Autodock Vina was shown in Table 4.

The relative positions of the four compounds docked to the P-gp are shown in Figure 4 and Figure 5. 

## 3. Discussion

*Anemarrhenae rhizoma*, particularly its saponins and flavonoids, has been demonstrated to have pharmacological functions against chronic myelogenous leukemia and multi-drug resistance tumors [27,28], and some of the function is related to the functional effects on P-gp. However, there is little information in the literature on the interaction between P-gp and the effective constituents from AR. 

In this study, an MDCK-MDR1 monolayer cell model was established [29]. In the permeability assay, the calculated P_e_ value of fluorescein sodium was (0.78 ± 0.02) × 10^−6^ cm/s in 120 min, which is in accordance with the international standard reference value for the definition of difficult drug absorption (*P_app_* < 1 × 10^−6^ cm/s) [30,31]. The results demonstrated the stability and reliability of the MDCK-MDR1 monolayer cell model.

The results of bidirectional transport of MDCK-MDR1 usually demonstrate the effect of P-gp on compound transport through the cell membrane. The bidirectional transport results of seven effective constituents of AR were conducted in the MDCK-MDR1 cell model in this study. The efflux ratio, which is the comparison between results obtained for apical–basolateral (A→B) and basolateral–apical (B→A) directions, may offer an idea regarding potential substrates for P-gp [32]. The efflux ratio of TAIII, TBII, MGF, and BHI were greater than 2 in the MDCK-MDR1 cell model, identified as P-gp substrates with the result of bidirectional transport experiments co-administration of verapamil. Moreover, the efflux ratio of IMGF, NMGF, and SSG were less than 2 in the MDCK-MDR1 cell model, indicating that transmembrane transport of IMGF, NMGF, and SSG can be regarded as passive transport.

Interestingly, the transport results of seven components in the herbal extract of AR were not consistent with those in monomers’ form. The P_app (AP-BL)_ values of both TAIII and TBII increased, and their absorption characteristics improved from moderate to high absorption, while the efflux of TAIII and TBII from the extract of AR was significantly decreased in MDCK-MDR1 cells. Considering that only TAIII has been shown to have the function of inhibiting P-gp [27] and that there are also significant changes in TAIII transport, the above results suggest that these seven constituents could be potential P-gp inhibitors. The increase or decrease in P_app (AP-BL)_, P_app (BL-AP),_ and ER values of other efficient constituents also suggest that the administration of AR extract may lead to synergy in the absorption of multiple effective constituents.

Rhodamine 123 was usually used to detect the effects of natural products on the activity of P-gp [33,34] and can be used to characterize the interplay between test compounds and P-gp to identify drugs as P-gp substrates or inhibitors [35]. A series of Rh-123 transport assays were conducted to investigate the inhibition of P-gp function by the seven efficient constituents in AR. The inhibition or facilitation of P-gp by each effective constituent of AR was investigated using the Rh-123 in combination with the active ingredient as well as the positive inhibitor valspodar. The results showed that TAIII, TBII, MGF, IMGF, NMGF, and BHI significantly reduced the ER of Rh123 in cells (*p* < 0.05) and may have an inhibitory effect on P-gp function. IMGF is functionally equivalent to the third-generation P-gp inhibitor valspodar in decreasing Rh-123 efflux.

Our research illustrates that TAIII, TBII, MGF, and BHI are both substrates and inhibitors of P-gp. So, molecular dockings were performed to investigate the binding of the four compounds with P-gp. According to a relevant study [36], some P-gp inhibitors can perform as substrates at low concentrations but interfere with the early steps of the peristaltic extrusion mechanism at higher concentrations. The binding sites for many P-gp inhibitors are the cavity and the access tunnel of P-gp, like tariquidar and elacridar [36,37]. In this study, using of P-gp as a transporter in molecular docking revealed that TAIII, TBII, MGF, and BHI bind roughly, overlapping the binding sites for elacridar, the substrates, and inhibitors of P-gp. We also performed the binding of the four compounds with the nucleotide-binding domain of P-gp. Verapamil is a typical inhibitor combined with a nucleotide-binding domain [38]. MGF and BHI binding sites roughly overlap with their binding site.

There is a wide range of P-gp inhibitors, including anthracyclines, peptide antibiotics, alkaloids, and steroid hormones. The chemical structure and properties of the P-gp substrates differ widely, so the results obtained from a single substrate may not be accurate. P-gp inhibitors have been developed for nearly 40 years, and the fourth generation of inhibitors is currently expected to be screened from natural products [39]. The results of this study indicate that TAIII, TBII, MGF, IMGF, NMGF, and BHI have the potential to be developed as P-gp inhibitors and could serve as a reference for the screening of P-gp inhibitors.

## 4. Materials and Methods

### 4.1. Chemicals

Timosaponin AIII (ST11240120), timosaponin BII (ST07210120), sarsasapogenin (ST01110120), mangiferin (ST00350120), neomangiferin (ST00360120), isomangiferin (ST14870105) and baohuoside I (ST07960120) (purity > 98%) were purchased from Nature-standard Inc. (Shanghai, China). Rhodamine-123 was obtained from Sigma Inc. (Marlborough, MA, USA). Verapamil was purchased from Dalian Meilun Biotechnology Co. (Dalian, China). Carbamazepine was bought from Shanghai ZZBIO Co. (Shanghai, China). Puerarin was purchased from Shanghai Yiyan bio-technology Co. (Shanghai, China). Valspodar was obtained from Delta bio-technology Co. (Nanchang, China). Dulbecco’s Modified Eagle’s Medium (DMEM) medium, fetal bovine serum (FBS), phosphate-buffered saline (PBS), and 0.25% Trypsin were bought from Corning Inc. (Corning, NY, USA). Penicillin/streptomycin was purchased from Gibco Inc. (Grand Island, NY, USA). Purified water was taken from a Milli-Q system (Millipore Corp., Billerica, MA, USA). All solvents used were HPLC grade, and all chemicals were analytical grade. Transwell polycarbonate inserts (12-well, 0.4 μm pore size) were obtained from Costar Inc. (Washington, DC, USA).

### 4.2. Cell Lines and Culture Conditions

MDCK-MDR1 cell lines were generously provided by Dr. Su Zeng (Zhejiang University). The cell line highly expresses the human ABCB1 gene stably. Cells were cultured in DMEM supplemented with 10% FBS, 1% Penicillin streptomycin solution at 37 °C, 5%CO_2,_ and 95% humidity. Cells were plated on Corning Transwell polycarbonate membrane cell culture inserts at a density of 5 × 10^5^ cells/well for 0.4 μm pore inserts, respectively. Cells were plated for seven days, and the confluency was about 95%. TEER values were measured every other day in preparation for subsequent assays.

### 4.3. Plant Materials

*Anemarrhenae rhizoma* was sourced from Shanghai Lei Yun Shang Pharmaceutical Co. (Shanghai, China). After botanical identification through comparative macroscopic and microscopic studies by Prof. Meili Guo (Pharmacognosy, Naval Medical University, China), plant exsiccates (No. 20200801) were deposited in the Herbarium of Department of Pharmaceutical Analysis, Naval Medical University, at the city of Shanghai, China.

### 4.4. Preparation of AR Extract

AR extract was prepared using heat-reflux extraction. A 50 g powder sample of AR was extracted with 400 mL 95% ethanol solution for 90 min at 95 °C twice, which were then combined, filtered, and concentrated to 50 mL. The rest residue also was extracted with 400 mL 50% ethanol aqueous solution for 90 min at 95 °C twice, which was then combined, filtered, and concentrated to 50 mL. The filtrates were combined, and the final extract was evaporated to approximately 50 mL.

### 4.5. Cytotoxicity Assay

The in vitro cell viability was determined by CCK-8 assays to interpret the non-cytotoxic concentration of seven compounds. 5 × 10^3^ MDCK-MDR1 cells per well were inoculated into 96-well plates. After the cells were incubated for 24 h at 5% CO_2_ and 37 °C, different concentrations of TAIII, TBII, SSG, MGF, NMGF, IMGF, BHI (0.1–48 µM/mL) or AR extract (0.4–1.0 μg/mL) were applied into wells, respectively. Cells were incubated for 24 h before adding 10 µL of CCK-8 solution per well for 2 h. Absorbance (450 nm) was measured using a Hybrid Microplate Reader (Synergy 4, BioTek, Winooski, VT, USA). 

### 4.6. Bidirectional Transport Assays

#### 4.6.1. Substrate Assays for Seven Components in AR

MDCK-MDR1 cells with TEER values higher than 450 Ω/cm^2^ and P_app_ values of Lucifer yellow less than 1 × 10^6^ cm/s were used for the experiments. The HBSS containing different component (2.0 µM TAIII, 1.6 μM TBII, 4.7 μM MGF, 4.7 μM IMGF, 3.4 μM NMGF 1.2 μM SSG or 3.9 μM BHI) were prepared. Before experiments, cell monolayers were washed twice with HBSS and equilibrated with HBSS for 30 min at 37 °C. HBSS containing different components was added to either the apical or basolateral chamber (donor chamber), and blank HBSS was added to the opposite chamber (receiver chamber). The total volume is 0.5 mL in the apical chamber and 1.5 mL in the basolateral chamber. Aliquots (100 μL) were taken from the acceptor chamber every 30 min for 2 h, and the same volume of HBSS was replenished. 

#### 4.6.2. Substrate Assays for AR Extract

The HBSS containing 10 mg/mL AR extract was prepared. Before experiments, cell monolayers were washed twice with HBSS and equilibrated with HBSS for 30 min at 37 °C. HBSS containing AR extract was added to either the apical or basolateral chamber (donor chamber), and blank HBSS was added to the opposite chamber (receiver chamber). The total volume is 0.5 mL in the apical chamber and 1.5 mL in the basolateral chamber. Aliquots (100 μL) were taken from the acceptor chamber every 30 min for 2 h, and the same volume of HBSS was replenished. 

### 4.7. Sample Pretreatment and LC-MS/MS Analysis

The transport samples were prepared using solid-phase extraction (SPE). Bond Elut C18 cartridge (Agilent, Santa Clara, CA, USA, 100 mg, 1 mL, 100/pk) was preconditioned with 1 mL methanol and 1 mL deionized water. Both 100 μL of bidirectional assay sample solution and 20 μL of IS solution was loaded onto the SPE cartridge, and the eluent that passed through the SPE cartridge was discarded. After washing with 1 mL of deionized water, 1 mL of the eluent solvent was introduced into the SPE cartridge for collection. The target solutions were concentrated by nitrogen blowing and re-dissolved. The amount of seven constituents of AR was determined by LC-MS/MS. 

The LC-QQQ-MS system consisted of an HPLC system (Agilent, USA) coupled with a triple quadrupole MS system with an electrospray ionization (ESI) source (Agilent, USA). Qualitative Analysis 10.0 and Quantitative Analysis 10.2 software (Agilent, USA) was used for qualitative and quantitative analysis. 

A C_18_ column (2.1 × 50 mm, 2.7 μm, Agilent, USA) was used in the system. 0.1% formic acid in water (A) and acetonitrile (B) was the mobile phase for gradient elution which was as follows: 0–2 min, 90% A: 2–3 min, 90%–10% A; 3–10 min, 10% A. The flow rate was set as 0.30 mL/min, and the temperature was set as 30 °C. The injection volume was 3 μL.

Mass spectrometry was performed in positive and negative modes in multiple reaction monitoring (MRM) modes. The source parameters were set as follows: N1 gas was 20 psi, N2 gas was 40 psi, the capillary voltage was 4000 V for positive mode and −4000V for negative mode, and the gas temperature was 350 °C. MS parameters of seven components and ISs are illustrated in Table 5.

### 4.8. Bidirectional Transport Assays of Rh-123

The HBSS containing 10 μM Rh-123 and different AR components as described in the above Section 4.6.1 were prepared. Additionally, The HBSS containing 10 μM Rh-123 and 0.625 μM valspodar were prepared as the positive control. Before experiments, cell monolayers were washed twice with HBSS and equilibrated with HBSS for 30 min at 37 °C. HBSS containing AR component and Rh-123 was added to either the apical or basolateral chamber (donor chamber), and HBSS containing AR component was added to the opposite chamber (receiver chamber). The total volume is 0.5 mL in the apical chamber and 1.5 mL in the basolateral chamber. Aliquots (100 μL) were taken from the acceptor chamber at 15 min, 30 min, 60 min, 90 min, and 120 min, and the same volume of blank HBSS was replenished. The positive control group was treated as above.

### 4.9. Molecular Docking

In silico calculations to invest the gate binding of TAIII, TBII, MGF, and BHI to P-gp used the P-glycoprotein at 3.4 A resolution PDB ID 4Q9H. Coordinates and partial charges for TAIII, TBII, MGF, BHI, Elacridar, and Verapamil were from NIH PubChem CID 15953793, 122173202, 5281647, 5488822, 119373, and 2520. MGLTools and Autodock Vina packages were used to calculate partial charges for receptor (P-glycoprotein) atoms and to perform docking, respectively. The transmembrane cavities were investigated in the docking search for the above four compounds. 

Semiflexible molecular docking calculations were performed using Autodock Vina with compounds having assigned Gasteiger charges. Potential maps were calculated in a 40 × 40 × 40 grid box, centered at x, y, z = (73.3, 40.7, 47.8) Å for TAIII, TBII, MGF, BHI and Verapamil in the transmembrane domain and centered at x, y, z = (68.7, 1.5, 3.3) Å for TAIII, TBII, MGF, BHI, and Elacridar in the nucleotide-binding domain. Nine runs have been performed with the total number of individuals in a generation, with maximum exhaustiveness set to 10. The resulting poses were clustered with a 3.0 Å tolerance, and the hydrogen bond pattern was analyzed with the AutodockTools scripts. After docking, the best-scored interactions between the ligands and the P-gp were visualized, and PyMOL was used to visualize the docking details.

### 4.10. Data Analysis

The apparent permeability (P_app_) of each substrate across MDCK-MDR1 cell monolayers in both the basolateral to apical (B > A) and apical to basolateral (A > B) directions were calculated using the following formula: (1)Papp=VaC0×A×ΔQΔt
where V_a_ is the volume of the acceptor chamber (B > A: 0.5 mL; A > B: 1.5 mL), C_0_ is the initial concentration of substrate in the donor chamber, A is the monolayer growth surface area, ΔQΔt is the rate of substrate transport. The efflux rate was calculated using the following formula:(2)ER=Papp(BL−AP)Papp(AP−BL)
where P_app (BL-AP)_ is the apparent permeability of the basolateral chamber to the apical chamber and P_app (AP-BL)_ is the apparent permeability of the apical chamber to the basolateral chamber. 

SPSS 23.0 was used to analyze the Analysis of Variance (ANOVA). Differences between groups were considered statistically significant for *p* values < 0.05. The statistical tests used were appropriately identified in the figure legends.

## 5. Conclusions

The present study investigated the interaction between the biologically active components in AR and P-gp. The bidirectional transport results showed that TAIII, TBII, MGF, and BHI were the substrates of P-gp. The Rh-123 transport results showed that TAIII, TBII, MGF, NMGF, IMGF, and BHI inhibit P-gp to varying degrees. An in silico molecular docking model confirmed the compounds’ binding with P-gp, showing binding to both the central binding cavity and the nucleotide-binding domain. However, the mechanism by which these compounds inhibit P-gp still needs to be verified by further experiments. The results of our study improved the comprehension of the absorption process of *Anemarrhenae rhizoma* and its effective components and provided a theoretical basis for its clinical applications.

## Figures and Tables

**Figure 1 molecules-27-08556-f001:**
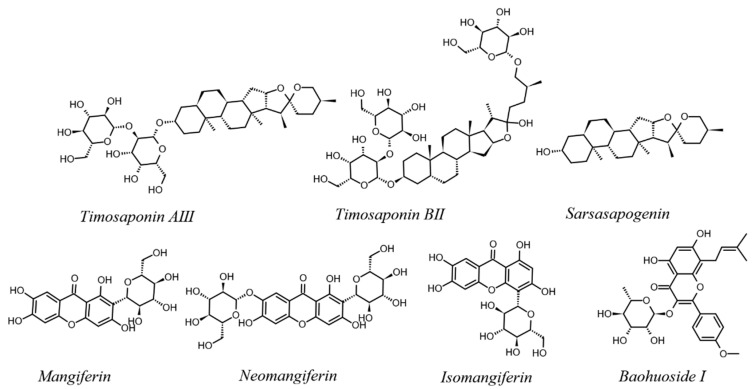
Chemical structures of seven compounds in AR.

**Figure 2 molecules-27-08556-f002:**
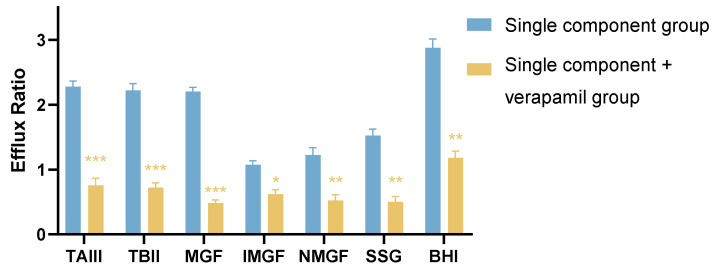
Effect of verapamil on the efflux ratio of timosaponin A III (TAIII), timosaponin B II (TBII), mangiferin (MGF), neomangiferin (NMGF), isomangiferin (IMGF), sarsasapogenin (SSG), and baohuoside I (BHI). Data are expressed as mean ± SD (n = 3). * *p* < 0.05, ** *p* < 0.01, *** *p* < 0.001 compared to single component group.

**Figure 3 molecules-27-08556-f003:**
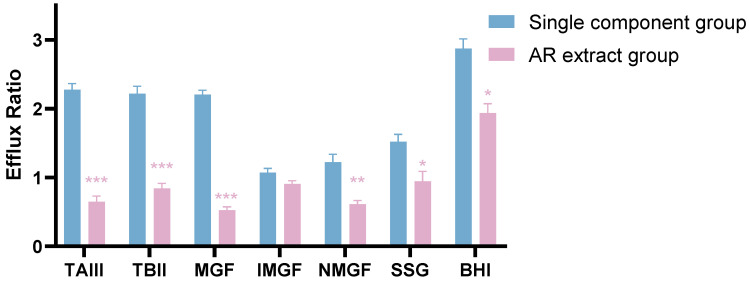
The efflux ratio of seven components in the administration of AR extract and monomers. Data are expressed as mean ± SD (n = 3). * *p* < 0.05, ** *p* < 0.01, *** *p* < 0.001 compared to Single component group.

**Figure 4 molecules-27-08556-f004:**
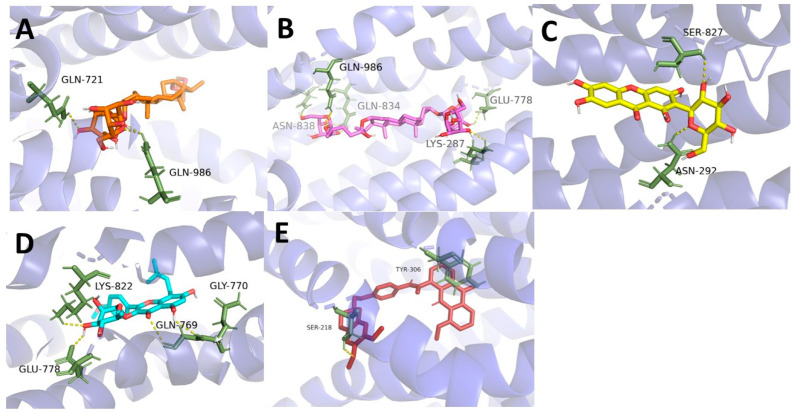
Binding interactions between the AR components and P-gp (Translucent purple) binding site. (**A**). TAIII (orange); (**B**). TBII (magenta); (**C**). MGF (yellow); (**D**). BHI (blue); (**E**). Elacridar (red). H-bonds, yellow dashed lines, Å.

**Figure 5 molecules-27-08556-f005:**
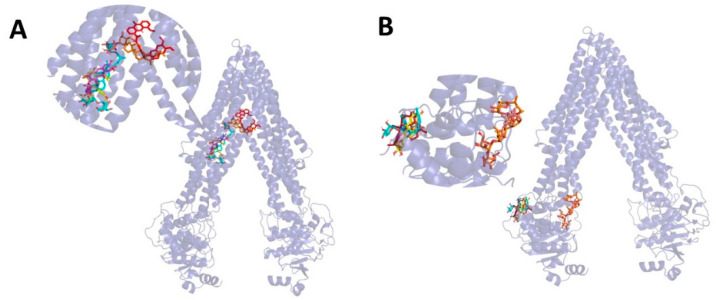
(**A**). P-gp showing inhibitor binding cavities and docked TAIII (orange), TBII (magenta), MGF (yellow), BHI (blue), and Elacridar (red). P-gp is shown in cartoon representation with slate purple. (**B**). P-gp showing nucleotide-binding domain and docked TAIII (orange), TBII (magenta), MGF (yellow), BHI (blue), and Verapamil (red). P-gp is shown in cartoon representation with slate purple.

**Table 1 molecules-27-08556-t001:** Effect of verapamil on the ER and Papp of timosaponin A III (TAIII), timosaponin B II (TBII), mangiferin (MGF), neomangiferin (NMGF), isomangiferin (IMGF), sarsasapogenin (SSG), and baohuoside I (BHI). (n = 3).

	Single Component Group			Single Component + Verapamil Group		
Compound	Papp (AP-BL) (×10^−8^ cm/s)	Papp (BL-AP) (×10^−8^ cm/s)	ER	Papp (AP-BL) (×10^−8^ cm/s)	Papp (BL-AP) (×10^−8^ cm/s)	ER
TAIII	95.54	215.81	2.26	24.36	21.55	0.88 ***
TBII	71.87	166.58	2.32	87.37	56.5	0.65 ***
MGF	60.6	134.98	2.23	63.03	33.41	0.53 **
IMGF	6.51	7.02	1.08	25.9	14.25	0.55 *
NMGF	12.06	13.65	1.13	52.89	27.38	0.52 **
SSG	3.74	5.66	1.51	9.04	3.74	0.41 **
BHI	23.98	72.04	3.00	84.8	91.88	1.08 **

* *p* < 0.05, ** *p*< 0.01, *** *p*< 0.001 compared to single component group.

**Table 2 molecules-27-08556-t002:** The ER and Papp of seven components in AR extract (n = 3).

Compound	Papp (AP-BL) (×10^−8^ cm/s)	Papp (BL-AP) (×10^−8^ cm/s)	ER	
			AR Extract Group	Single Component Group
TAIII	305.29	193.73	0.63 ***	2.26
TBII	181.30	150.25	0.83 ***	2.32
MGF	58.64	28.25	0.48 ***	2.23
IMGF	5.46	4.95	0.91	1.08
NMGF	9.76	6.54	0.67 **	1.13
SSG	3.98	3.47	0.87 **	1.51
BHI	15.51	29.78	1.92 *	3.00

* *p* < 0.05, ** *p* < 0.01, *** *p* < 0.001, compared to Single component group.

**Table 3 molecules-27-08556-t003:** The efflux ratio and Papp of rhodamine-123 in combination with seven constituents of AR and valspodar (n = 3).

Compound	Papp (AP-BL) (×10^−6^ cm/s)	Papp (BL-AP) (×10^−6^ cm/s)	ER
Control	1.89	4.65	2.46
Valspodar	1.94	2.02	1.04 ***
Timosaponin AIII	1.78	2.17	1.22 **
Timosaponin BII	1.85	2.34	1.27 **
Mangiferin	1.88	2.36	1.25 **
Isomangiferin	1.78	1.93	1.09 ***
Neomangiferin	1.92	2.51	1.31 **
Sarsasapogenin	1.45	2.80	1.92
Baohuoside I	1.73	2.53	1.47 *

* *p* < 0.05, ** *p* < 0.01, *** *p* < 0.001 compared to Control group.

**Table 4 molecules-27-08556-t004:** Binding free energy and involved key residues for TAIII, TBII, MGF, BHI, elacridar, and verapamil in the predicted binding sites of P-gp.

Compounds	Transmembrane Domain		Nucleotide-Binding Domain	
	Key Residue	Binding Free Energy (Kcal/mol)	Key Residue	Binding Free Energy (Kcal/mol)
Timosaponin AIII	GLN-721, GLN-986	−8.7	ARG-258, THR-259, GLN-1114, ILE-1226	−8.3
Timosaponin BII	GLN-986, ASN-838, GLN-834, GLU-778, LYS-287	−8.6	ILE-257, ILE-261, THR-259, ASP-1196	−7.9
Mangiferin	SER-827, ASN-292	−7.7	GLN-1108, ARD-1188	−7.8
Baohuoside I	GLU-778, LYS-822, GLN-769, GLY-770	−10.1	LYS-1023, ASP-1131, ASN-1132	−8.8
Elacridar	SER-218, TYR-306	−9.6	/	/
Verapamil	/	/	ARG-1188, GLN-1108, LYS-1023	−13.4

**Table 5 molecules-27-08556-t005:** MS parameter of seven components of AR and ISs.

ESI Mode	Compound	MRM Transition (Precursor-Product)	Fragmentor (V)	CID (eV)
Pos	Neomangiferin	607.0–426.7	140	34
	Baohuoside I	537.0–391.2	150	17
	Timosaponin BII	943.6–925.6	150	40
	Sarsasapogenin	417.0–273.2	120	15
	Carbamazepine (IS)	236.9–193.9	120	20
Neg	Isomangiferin	421.0–331.1	135	20
	Timosaponin AIII	739.7–577.2	120	33
	Mangiferin	421.0–331.1	135	20
	Puerarin (IS)	415.0–267.0	140	25

## Data Availability

Not applicable.

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
