# Peer review of "In Vitro Evaluation of the Interaction of Seven Biologically Active Components in *Anemarrhenae rhizoma* with P-gp"

_molecules, 2022, doi:10.3390/molecules27238556_

Round 1

Reviewer 1 Report

The manuscript is well-written and the findings are new and interesting. However, there is a need for English editing of the manuscript for minor grammar and spacing errors. For example, in line 341, the sentence started with "And the Rh-123 transport---" which is not grammatically correct. Also, there are several spacing errors like 50g, 100mg, 1mL, 60min, etc. There should be a space between the numbers and their corresponding units.

In Tables 1, 2 and 3, the units written as 10-6 or 10-8 cm/s should be corrected to make -6 or -8 appear as superscripts.

Author Response

Response to Reviewer 1 Comments

Point 1: The manuscript is well-written and the findings are new and interesting. However, there is a need for English editing of the manuscript for minor grammar and spacing errors. For example, in line 341, the sentence started with "And the Rh-123 transport---" which is not grammatically correct. Also, there are several spacing errors like 50g, 100mg, 1mL, 60min, etc. There should be a space between the numbers and their corresponding units.

Response 1: We are very grateful to you for reviewing the paper so carefully. We have carefully considered the suggestion of Reviewer and make some changes. We corrected the grammar errors and added a space between the numbers and their corresponding units(except for % and ℃)

Point 2: In Tables 1, 2 and 3, the units written as 10-6 or 10-8 cm/s should be corrected to make -6 or -8 appear as superscripts.

Response 2: Thank you for careful checking. We corrected the 10-6 or 10-8 cm/s as 10-6 or 10-8 cm/s in tables.

Reviewer 2 Report

The manuscript of Hong and colleagues concern an evaluation of the interaction of seven effective compounds of a dried rhizome Anemarrhenae rhizoma (AR) with the P-gp

The manuscript was developed with a sound rationale and wet-lab outcomes are corroborated by in-silico outcomes. 

Even though the manuscript deserves to be published, there are significant issues that should be addressed:

- The introduction is poor in terms of references regarding the use of natural products as P-gp inhibitors and/or modulators (just to cite someone: Molecules 2017, 22(6): 871. Pharmaceuticals 202114(2): 111)

- The P-gp protein has an ATP binding domain (NBP) that is responsible for the opening/closing of the pump. The authors just evaluated the potential binding with the DBP and did not use any reference compound (for instance Verapamil) to compare it with the AR compounds (please read, compare and cite Pharmaceuticals 202215(3): 356)

- The authors should evaluate the P-gp ATPase activity to determine if their compounds could influence the P-gp activity in this way

Author Response

Response to Reviewer 2 Comments

Point 1: The introduction is poor in terms of references regarding the use of natural products as P-gp inhibitors and/or modulators (just to cite someone: Molecule 2017, 22(6): 871. Pharmaceuticals 2021, 14(2): 111)

Response 1: We are very grateful to you for reviewing the paper so carefully. We appreciate it very much for this good suggestion. We searched and cited the relevant literatures, i.e. Refs 20,21,22,23, about the natural products as P-gp inhibitors or modulators.

Point 2: The P-gp protein has an ATP binding domain (NBP) that is responsible for the opening/closing of the pump. The authors just evaluated the potential binding with the DBP and did not use any reference compound (for instance Verapamil) to compare it with the AR compounds (please read, compare and cite Pharmaceuticals 2022, 15(3): 356)

Response 2: Thank you for reading and reviewing our manuscript. We selected two typical P-gp inhibitors with different binding sites, elacridar (for the transmembrane domain) and verapamil (for the nucleotide-binding domain). Four potential P-gp inhibitors were compared with their binding sites to elacridar and verapamil aiming to make the molecular docking results more convincing.

Point 3: The authors should evaluate the P-gp ATPase activity to determine if their compounds could influence the P-gp activity in this way

Response 3: Thank you for your good advice. We understand that P-gp ATPase activity assay may better reveal the compounds’ influence on P-gp activity. However, in the present study, we mainly focused on the interaction of the biologically active components and P-gp, especially that TAIII, TBII, MGF, and BHI are both substrates and inhibitors of P-gp. The results of Rh-123 transport assays were sufficient to draw a conclusion that the above biologically active components can inhibit P-gp to varying degrees (for reference: Molecules 2015, 20, 2931-2948 and Food Res Int 2018 Jan;103:110-120). We are willing to evaluate the P-gp ATPase activity in further studies.

Reviewer 3 Report

The manuscript entitled "In vitro evaluation of the interaction of seven effective components in Anemarrhenae Rhizoma with P-gp" is interesting and represents a novel study ot transport mechanisms of biologically active substance for intracellular delivery. I believe the manuscript will be highly appreciated by the readers of Molecules journal. I have few recommendations and remarks:

1. I would recommend using "biologically active substances" instead of "effective"in the title.

2. Abstract: the autors write "The efficacy and pharmacokinetics of Anemarrhenae rhizoma"... Actually, that would refer to the components of the raw material, not for the rhizoma. The sentence should be revosed.

3. Multiple abbreviations were used in the abstract without being introduced earlier in the text.

4. Line 52 - and anti-disease Alzheimer's properties - please correct this phrase, it makes no sense.

5. Figure 2 and table 1 - define the tested substances

6. Line 117 - the author use the term AR extract. This is the first time to mention using extract but not any other type of plant material, so this should be mentioned earlier, in the indtroduction.

7. Whi is section 2.5 Data analysis in italic?

8. Line 172 - is it just one saponin and one flavonoid? Revise the sentence.

9. Section 4.1. - Subsection title does not correspond to the content.

10. Section 4.4 - what was the method for extraction? It should be thoroughly described. I have some concernes using ethanol 95% at 95 degrees C.

11. Line 272 - The author mention TEER values but confluency of the monolayer formed was never discussed before. They should describe it in the methodology section.

12. Why is chapter 4.9 in italic?

Author Response

Response to Reviewer 3 Comments

Point 1: I would recommend using "biologically active substances" instead of "effective" in the title.

Response 1: We are very grateful to you for reviewing the paper so carefully. We have changed the title to “In vitro evaluation of the interaction of seven biologically active components in Anemarrhenae Rhizoma with P-gp”

Point 2: Abstract: the authors write "The efficacy and pharmacokinetics of Anemarrhenae rhizoma"... Actually, that would refer to the components of the raw material, not for the rhizoma. The sentence should be revosed.

Response 2: We are very grateful to you for reviewing the paper so carefully. We changed line 14 to " The efficacy and pharmacokinetics of the biologically active components in Anemarrhenae rhizoma (AR) ...." to make the sentence fit better with the experiment.

Point 3: Multiple abbreviations were used in the abstract without being introduced earlier in the text.

Response 3: We introduced the P-glycoprotein and Rhodamine-123 in the abstract before the abbreviations.

Point 4: Line 52 - and anti-disease Alzheimer's properties - please correct this phrase, it makes no sense.

Response 4: We correct the phrase as “and ameliorate learning deficits in mice”

Point 5: Figure 2 and table 1 - define the tested substances

Response 5: With reference to other literature, we define the abbreviation of biologically active components of its first appearance in the figure and table.

Point 6: Line 117 - the author uses the term AR extract. This is the first time to mention using extract but not any other type of plant material, so this should be mentioned earlier, in the introduction.

Response 6: We have mentioned the AR extract in introduce, line 95.

Point 7: Why is section 2.5 Data analysis in italic?

Response 7: We’ve change it in Palatino Linotype.

Point 8: Line 172 - is it just one saponin and one flavonoid? Revise the sentence.

Response 8: We correct the phrase as” Anemarrhenae rhizoma, particularly its saponins and flavonoids…”

Point 9: Section 4.1. - Subsection title does not correspond to the content.

Response 9: We are very grateful to you for reviewing. We have corrected the subsection title as ”Chemicals”

Point 10: Section 4.4 - what was the method for extraction? It should be thoroughly described. I have some concernes using ethanol 95% at 95 degrees C.

Response 10: The method for extraction was heat-reflux extraction and the method has been described in section 4.4. Using 95% ethanol at 95 degree C is safe and efficient in the extraction of natural products.

Point 11: Line 272 - The author mention TEER values but confluency of the monolayer formed was never discussed before. They should describe it in the methodology section.

Response 11: We’ve described the confluency and measure of TEER values in section 4.2.

Point 12: Why is chapter 4.9 in italic?

Response 12: We’ve change it in Palatino Linotype.

Reviewer 4 Report

This manuscript aims to examine the transmembrane absorption of several compounds in Anemarrhenae rhizome and their interaction with P-gp. Please consider the following observations: Please modify the caption of subchapter 4.1 to “Chemicals” or “Reagents”. Please provide a more detailed Conclusions section. Please update the reference section with papers published in the last 5 years. More references should be added to this section.

Author Response

Response to Reviewer 4 Comments

Point 1: This manuscript aims to examine the transmembrane absorption of several compounds in Anemarrhenae rhizome and their interaction with P-gp. Please consider the following observations: Please modify the caption of subchapter 4.1 to “Chemicals” or “Reagents”. Please provide a more detailed Conclusions section. Please update the reference section with papers published in the last 5 years. More references should be added to this section.

Response 1: We are very grateful to you for reviewing the paper so carefully. We corrected the subsection title as ”Chemicals”. We have expanded the Conclusion section to make it more convincing and forward-looking. We have also added references, i.e. Refs 20, 21, 22, 23, 33, 34, 35 and 38, to make our articles more informative.

Round 2

Reviewer 2 Report

The authors have replied to all the issues I raised improving the manuscript that is now suitable to be published

I have no further comments